# Malignant Gastrointestinal Neuroectodermal Tumor: A Case Report and Literary Review for a Rare Differential Diagnosis

**Cinzia Baccaro** [1], **Noemi Zorzetti** [1,2,*], **Manuela Cuoghi** [1], **Adele Fornelli** [3], **Tania Franceschini** [3], **Sara Coluccelli** [4], **Vincenzo Cennamo** [5] and **Giuseppe Giovanni Navarra** [1]

1    Department of General Surgery, "A. Costa" Hospital—Alto Reno Terme, 40046 Bologna, Italy
2    Department of Surgical Sciences, La Sapienza University, 00185 Rome, Italy
3    Department of Pathological Anatomy, Maggiore "Carlo Alberto Pizzardi" Hospital, 40133 Bologna, Italy
4    Solid Tumor Molecular Pathology Laboratory, Department of Medical and Surgical Sciences (DIMEC), IRCCS Azienda Ospedaliero-Universitaria, 00144 Bologna, Italy
5    Department of Gastroenterology, Maggiore "Carlo Alberto Pizzardi" Hospital, 40133 Bologna, Italy
*    Correspondence: noemi.zorzetti@gmail.com

**Abstract:** Malignant gastrointestinal neuroectodermal tumor (GNET) is an infrequent soft-tissue sarcoma, formerly referred to as clear-cell sarcoma-like gastrointestinal tumor (CCSLGT) and frequently reported in the literature as clear-cell sarcoma of the gastrointestinal tract (CCS-GI); it is characterized by an absence of melanocytic differentiation and the presence of nontumoral osteoclast-like giant cells (OLGCs). The current study reports a case of a 79 year old woman admitted to the emergency department (ED) with symptoms of constipation and intestinal obstruction; a mass was found within the ileal wall necessitating of surgical approach. Immunohistochemically, tumor cells surprisingly had the hallmark of GNETs. Unfamiliarity with tumors with the features of GNETs can easily lead to a misdiagnosis by surgical pathologist. Therefore, comprehensive evaluation, including morphology and additional studies, is required for an appropriated diagnosis. Furthermore, without a high index of suspicion, there is actually no consensus on staging or treatment.

**Keywords:** malignant gastrointestinal neuroectodermal tumor; soft-tissue sarcoma; intestinal obstruction; ileum S-100 protein





## 1. Introduction

Malignant gastrointestinal neuroectodermal tumor (GNET) is an extremely rare and controversial entity [1–3], first described in 2003 by Zambrano et al. as a 'clear sarcoma-like tumor of the gastrointestinal tract'; in this study, they reported six cases of an osteoclast-rich tumor of the gastrointestinal tract with features resembling clear-cell sarcoma of soft parts [4], thus giving birth to the concept of a new class of tumors.

GNET shares some, but not all, the characteristics of clear-cell sarcoma (CCS) such as positivity for vimentin and S-100, or similar molecular translocation of *t(12;22) (q13;q12): EWS-ATF1*, and *t(2;22)(q32.3;q12): EWS-CREB1* fusion genes; however, the absence of other melanocytic-associated marker expression and the leak of melanocytic differentiation in electron microscopy analyses are helpful for differentiating GNET from CCS of soft parts. In addition, CCSs occur in the deep soft tissues of the extremities, trunk, or limb girdles, with a predilection near the tendon, fascia, or aponeuroses [4], while GNETs occur usually occur in the gastrointestinal tract [1,2]. These two differences, the first genetic and the second related to the site of occurrence, can be considered as cornerstones in the differential diagnosis.

In 2012, Stockman et al. [3] proposed to redesignate this tumor class as GNET instead of 'clear-cell sarcoma-like tumor of the gastrointestinal tract' (CCSLTGT), and this term has been increasingly accepted by pathologists [1–6]. Furthermore, GNET can be histologically misdiagnosed as another epithelioid and/or spindle cell neoplasm [7].

In 2019, Huang et al. reported, in a review of literature, the clinicopathological and cytogenetic features of 47 cases of GNETs; all the described cases arose within the abdominal cavity involving the small intestine, stomach, or colon, and patients often presented nonspecific tumor-related symptoms but only generic ones such as weight loss, abdominal pain with intestinal obstruction, or anemia. Histologically, tumor cells present with a solid, nested, pseudo-alveolar, or fascicular growth pattern, occasionally forming a pseudopapillary microcystic or rosette-like architecture [8].

## 2. Case Reports and Evolution

A 79 year old woman was admitted to ED in 'A. Costa Hospital' Alto Reno Terme, Bologna, Italy for constipation and abdominal pain; before hospital access, she reported some episodes of vomiting at home. The patient was independent in her daily activities and had no allergies. Her primary medical comorbidities were DMNID, dyslipidemia, hypertension, hypothyroidism, and a previous heart attack in cardiological follow-up.

Laboratory test reported a hemoglobin value of 10.5 g/dL (12.5 g/dL < n.v. < 17.2 g/dL) and C-reactive protein (CRP) 1.2 mg/dL (0 mg/dL < n.v. < 0.5 mg/dL).

A CT abdominal scan was performed in the suspect of intestinal obstruction showing a mass of about 30 mm involving the small bowel, complicated by an ileal–ileal invagination, and a right ovarian lesion of about 40 mm suspected as hamartoma (Figure 1a,b). The patient was, therefore, hospitalized in our surgical department and, considering the global good conditions, was initially treated with conservative therapy.

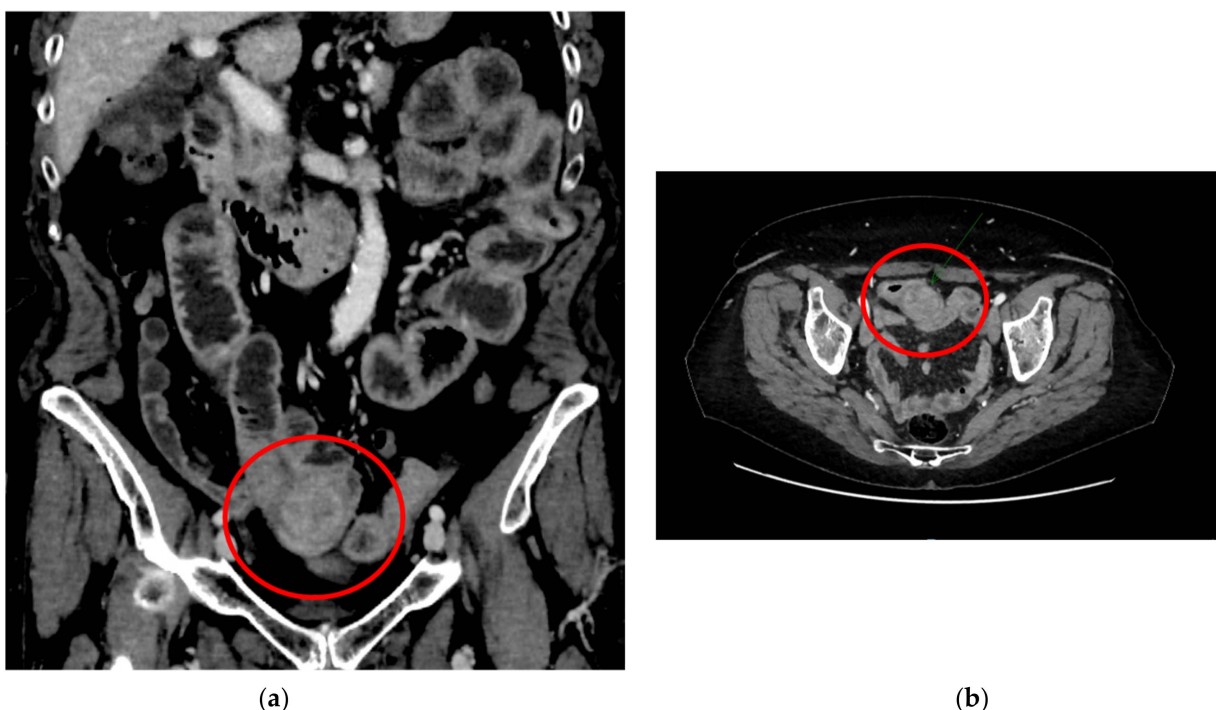

(**a**)                                                                 (**b**)

**Figure 1.** The circles show the mass involving the small bowel and the ileal–ileal invagination in the delayed phase: (**a**) sagittal plane; (**b**) transversal plane.

Subsequently, to further characterize the lesion, the patient underwent thoracic CT scan to complete the diagnostic staging that did not show any local lesion. Moreover, an abdominal MRI was also performed confirming the ileal mass and showing the ovarian tumor with the features of a teratoma (Figure 2). Lastly, a gynecological consultation was requested, which did not suggest any therapeutical indication for the teratoma lesion. Considering the site of the lesion, any kind of endoscopy could be performed to obtain preoperative biopsies.

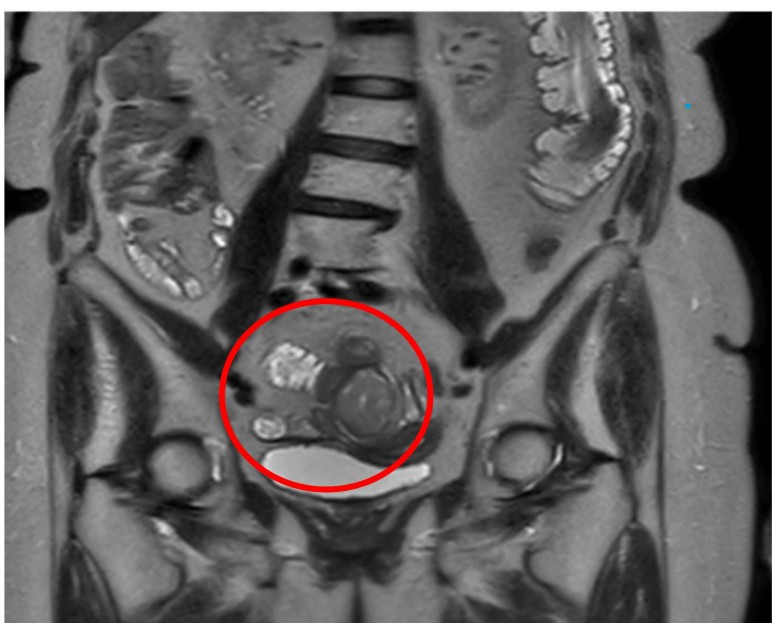

**Figure 2.** The circle shows the mass on MRI.

Afterward, once the clinical case was discussed at the multidisciplinary hospital gastrointestinal meeting, the patient underwent elective surgery; laparoscopic exploration of the abdominal cavity was performed, without identifying peritoneal and liver metastasis or ascites; moreover, laparoscopic peritoneal washing was collected. After a difficult laparoscopic tumor dissection and recognition of intestinal loops afferent to the lesion, three laparoscopic intestinal resections, including the intestinal mass (Figure 3), were performed with subsequent reconstruction of the intestinal tract with intracorporeal lateral-lateral mechanical anastomosis; bowel perfusion was controlled by indocyanine green fluorescence angiography before performing anastomosis, not identifying ischemic areas.

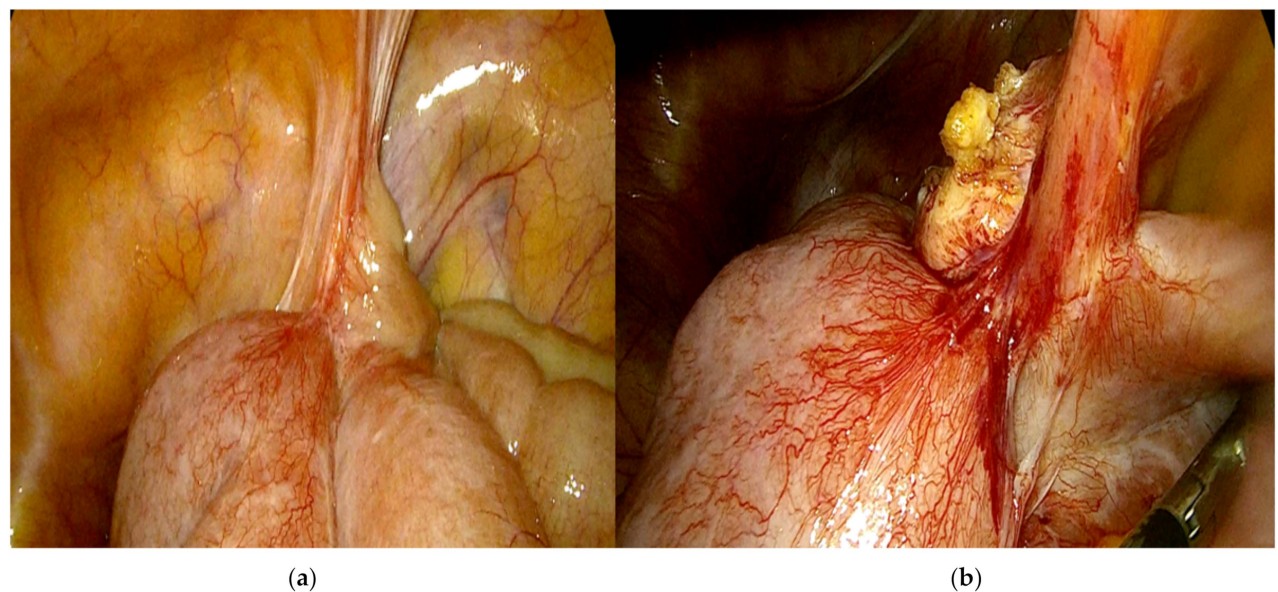

|  |  |
|:---:|:---:|
| (**a**) | (**b**) |

**Figure 3.** The ileal mass complicated by ileal–ileal invagination: laparoscopic view (**a**,**b**).

The patient's conditions improved dramatically after the treatment. The patient was discharged from hospital on the seventh postoperative day.

Pathological examination of the resected ileum confirmed the mass which invaded the ileal wall.

At histology, it was composed of spindle and round cells with a scarce eosinophilic and clear cytoplasm, as well as vesicular nuclei with small to medium-sized nucleoli. The cells were arranged in sheets and nests and showed a vaguely pseudo-alveolar architecture. Few osteoclast-like giant cells were also present (Figure 4).

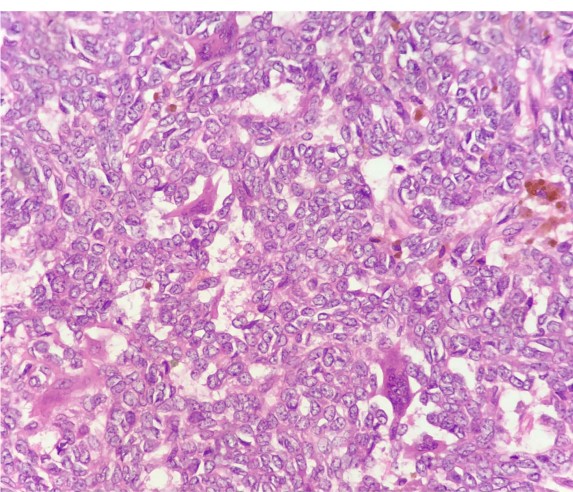

**Figure 4.** Few osteoclast-like giant cells are present within the neoplastic population (H&E 40×).

No areas of necrosis were present; the mitoses were 8/10 HPF. The proliferation index Ki67 was 18%; no vascular and neither perineural invasion was seen (Figure 5a,b).

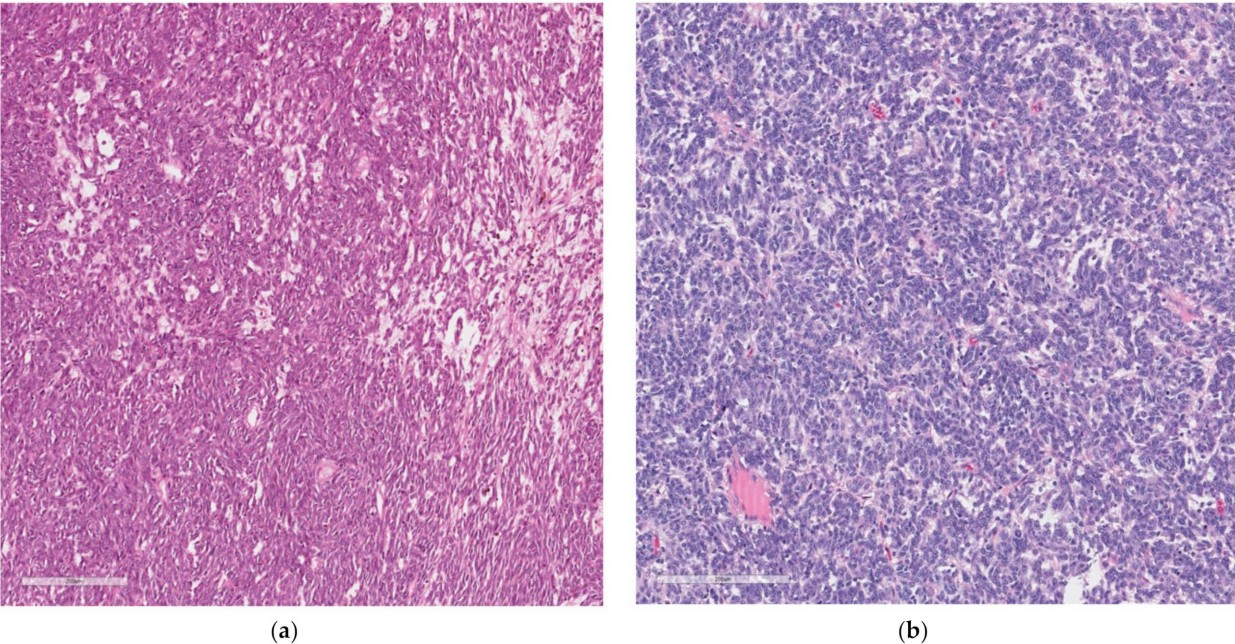

(**a**) (**b**)

**Figure 5.** Medium-power spindle (**a**) and medium-power epithelioid (**b**): the lesion is composed of spindle (**a**) and epithelioid cells (**b**) with a vaguely nested pattern (H&E, 200×).

Immunohistochemically, the neoplastic cells were positive for S-100 protein, synaptophysin, and SOX10, but negative for melanocytic (HMB45, Mart1, MitF) and epithelial markers (cytokeratin CAM 5.2, cytokeratin WS). In addition, CD117, DOG1, CD99, GATA3, alpha-inhibin, STAT6, PAX8, CDX2, chromogranin A, and smooth muscle markers (smooth muscle actin and desmin) were negative. Fluorescence in situ hybridization (FISH) analysis for *EWSR1* was performed using break-apart probes (EWSR1 Break Apart Probe, TITAN

FISH PROBES, OACP IE LTD, Phoenix House, Monahan Road, T12H1XY, Cork, Ireland) on paraffin-embedded sections (Figure 6).

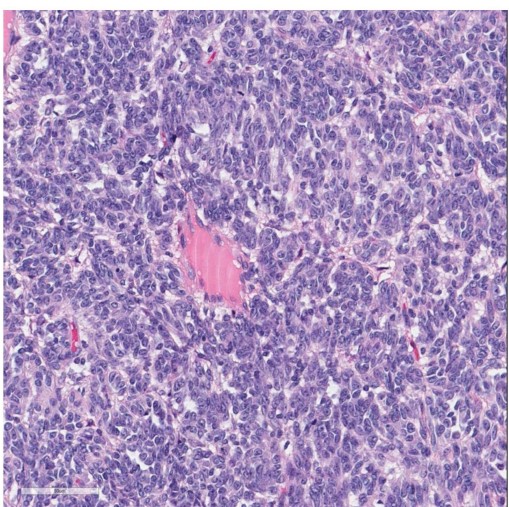

**Figure 6.** At high power, neoplastic cells show vesicular chromatin with small nucleoli (H&E, 400×).

The analysis revealed that 32.3% of neoplastic nuclei harbored the split of the *EWSR1* signal, indicative of a *22q12* rearrangement (Figure 7). That panel confirmed the diagnosis of GNET [8]. The patient subsequently performed an oncological evaluation, whose assessment led to the decision not to undergo any oncological therapy and required a close follow-up. The 6 month follow-up was uneventful without evidence of recurrent disease.

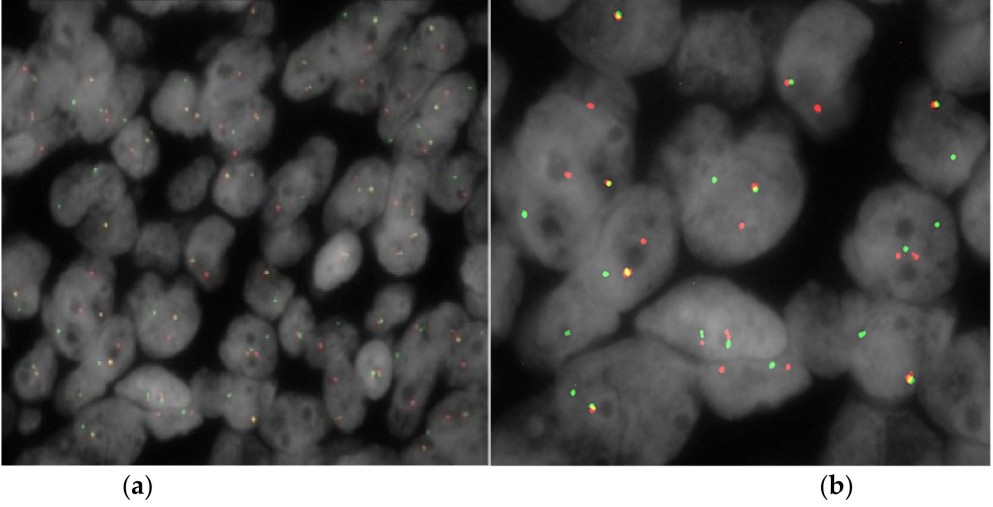

(**a**)         (**b**)

**Figure 7.** FISH analysis of the *EWSR1* (EWS RNA-binding protein 1) gene rearrangement status in a case of malignant gastrointestinal neuroectodermal tumor (GNET): (**a**) red color shows hybridization signals relative to the 5'-*EWSR1* gene, while green color shows hybridization signals relative to the 3'-*EWSR1* gene; (**b**) higher-magnification image of neoplastic nuclei with a gene fusion and at least one split of the red and green signals or a single red signal indicative of a chromosomal rearrangement in the EWSR1 gene region. Magnification: 100×.

## 3. Discussion

We performed research on PubMed with the following keywords: malignant gastrointestinal neuroectodermal tumor, soft-tissue tumors, and intestinal obstruction.

GNET is a rare neoplasm; since 2019, fewer than 60 cases have been described in the literature, most of them being case reports or small series studies [9]. Recently, Kandler et al. reported three new cases of metastatic GNET and reviewed the existing literature reaching

111 cases [10]. These tumors mostly occur in young to middle-aged adults (median age 35 years); there is no evident gender predominance, and tumors are usually located in the gastrointestinal tract [8]

Patients generally present no tumor-related specific symptoms, instead presenting only generic and ordinary ones such as abdominal pain, intestinal obstruction signs, and others related to the presence of an abdominal mass; moreover, at the time of diagnosis, some patients have metastatic disease.

Most cases have been described in the small intestine in addition to the stomach and the colon [1–4], following an aggressive clinical behavior with local recurrence and metastatic disease to lymph nodes or hematogenous spread early at the time of diagnosis [4–6].

The etiology is unknown; GNET is centered in the wall of the GI tract and may show either exophytic or endophytic growth. The tumor cells grow in solid sheets with pseudopapillary, alveolar, and nest formation [4]. Neoplastic cells are epithelioid and/or spindle with a variable amount of eosinophilic or clear cytoplasm. The nucleoli are inconspicuous, sometimes prominent, and basophilic. Osteoclast-like multinucleated giant cells are present [1,2]. Immunohistochemically, the neoplastic cells are strongly and diffusely positive for S-100 protein and SOX10, but negative for epithelial and melanocytic markers, as well as for CD117 and DOG1 antibodies.

A recent and remarkable analysis of 19 patients affected by GNET was proposed by Chang at al., describing the clinicopathological features and the treatment needed to provide an appropriate therapy. They found that the tumor was located in the small intestine in 11 patients, in the stomach in three cases, in the lower esophagus in one case, in the large intestine in two cases, in the anal canal in one case, and in the ileocecal junction in one case; this proves that GNET is located predominantly in gastrointestinal tract, being able to occur along its entire length. Only three cases received a correct initial diagnosis. Four patients showed lymph node metastasis at the time of diagnosis, while one patient showed liver metastases; one patient was lost to follow-up, and one case had unknown metastasis status and stage. During follow-up, seven patients developed liver metastases; this supports the thesis that GNET is an aggressive tumor. In a mean clinical follow-up of 29.7 months, two patients died, five patients were alive with tumor, eight patients were alive without tumor, and four patients were lost. Only three patients were treated with apatinib and anlotinib with clinical benefit, indicating that they might be effective for the treatment of advanced GNET. Lastly, two patients with liver metastases underwent surgery and radiofrequency ablation with benefit [9]. According to this paper, in contrast to the previous reported studies in the literature, the therapeutical strategy partially showed a better prognosis and probably achieved targeted treatment; in patients that show advanced GNET with hepatic metastases, the therapeutical strategy of tumor dissection and radiofrequency of the liver metastases can also be considered for a better prognosis at follow-up.

Kandle et al. reported three cases of GNET strongly relevant for the treatment of metastatic disease; the first case, initially diagnosed from a colonoscopy biopsy as a schwannoma of the sigma, underwent laparoscopy sigmoidectomy and laparoscopy biopsy for a deposit of the pelvic peritoneum found during the procedure. The pathology showed a GNET with a metastatic peritoneum; after a negative follow-up, a diagnostic laparoscopy was performed in order to assess the distribution of peritoneal disease and the potential for resectability. In the second case, a laparoscopic small bowel resection for an intestinal bowel obstruction was performed; the pathology demonstrated a malignant GNET with a clear resection margin but with lymphovascular invasion. No chemotherapy was performed; after 2 years, a CT scan revealed a 10 cm mass of the right lobe of the liver, and an ultrasound biopsy confirmed metastatic GNET. Palliative chemotherapy with three cycles of doxorubicin and olaratumab was administered without response. A portal vein embolization followed by right hemicolectomy and caudate lobe resection was performed. The pathology confirmed metastatic GNET. At close follow-up, there was no recurrence of disease. In the last case, an ileocolic resection was performed for iron-deficient anemia, and an abnormally thickened distal small bowel with lymphadenopathy was revealed by CT

scan in a patient with a history of thymic B-cell lymphoma treated with chemotherapy and chest radiation. The pathology confirmed the diagnosis of GNET; after 2 years of follow-up, a CT scan indicated multiple hepatic and osseus metastases in the spine. A liver biopsy was performed and confirmed metastatic GNET. After palliative spinal radiation treatment with significant reduction in bone pain, three cycles of doxorubicin were administered without response; the treatment was discontinued because of toxicity and intolerance. For progressive osseous metastases, palliative radiotherapy was performed; however, 36 months after diagnosis, the patient unfortunately died.

Kandler et al. conducted a detailed analysis of all 111 GNET cases described in the literature, highlighting the treatment and clinical outcome; they concluded that GNET is a biologically heterogenous disease that needs substantial pathologic expertise. Currently, surgery remains the main treatment modality in localized and metastatic GNET; for GNET with peritoneal metastasis, there are no studies that establishing the role of cytoreductive surgery and the feasibility of intraperitoneal chemotherapy [10].

GNET can be histologically misdiagnosed as another epithelioid and/or spindle cell neoplasm such as gastrointestinal stromal tumor (GIST), neuroendocrine tumor (NET), paraganglioma, metastatic melanoma, malignant peripheral nerve sheath tumor (MPNST), and synovial sarcoma (SS) by pathologists unaware of its existence; hence, differential diagnosis can be challenging [7].

In the literature, other epithelioid and/or spindle cell neoplasms are widely described, and all respective characteristics should be kept in mind to obtain a quicker and more accurate diagnosis (Table 1).

GISTs usually occur in middle-aged or older persons and can be asymptomatic. Occasionally, they can be found incidentally during surgery for other reasons or may present pain, bleeding, or signs of obstruction. They rarely show osteoclast-like multinucleated giant cells [11] but express CD117, DOG-1, and CD 34 antibodies; moreover, they are negative for S-100, as well as epithelial and melanocytic markers.

Metastatic melanomas need to be considered in the differential diagnosis because it is a great histological mimicker and shows diffuse S-100 protein immunoreactivity. The clinical history, along with a positive immunohistochemical positive reaction for other melanocytic markers such as HMB45, Mart1, and Mitf, should help to rule out this entity [12].

NETs are tumors that arise from cells of the endocrine and nervous systems; they most often occur in the intestine, where they are often called carcinoid tumors, but they are also found in the pancreas, lung, and rest of the body. They can be confused with GNETs because they have a similar architecture, but neoplastic cells are very rarely spindle-shaped; at immunohistochemistry, in addition to synaptophysin positivity, they are positive for chromogranin and epithelial markers, and S-100 is constantly negative [12].

Paraganglioma, on the other hand, is a rare neuroendocrine tumor derived from the nonepithelial neural crest, arising from the adrenal medulla (pheochromocytoma) and extra-adrenal paraganglia. It is often intra-abdominally located, near the superior para-aortic area; it has been found in the gallbladder and hepatic biliary tree, as well as in unusual sites such as the lung and genitourinary tract [13,14]. According to the immunohistochemistry of classic forms, they show S-100 positivity in sustentacular cells which surround neoplastic cell nests; in addition, neoplastic cells, which are characteristically synaptophysin-positive, show GATA 3 and alpha-inhibin expression [13–15].

MPNST is a rare tumor of the lining of the nerves, often occurring in the deep tissue of the arms, legs, and trunk; it often appears in patients with a history of neurofibroma and is characterized by a proliferation of spindle cells with alternative hypercellular and hypocellular areas, often showing weak and focal S-100 protein reactivity and potential positivity for cytokeratin antibodies.

Then, monophasic synovial sarcoma can arise in the GI tract, and the distinction from GNET can be difficult based on morphology alone [16]. Synovial sarcoma is a tumor of the soft tissue often found in the arm, leg, or feet and near joints such as the wrist or ankle; it occurs in young people under the age of 30, characterized by epithelial marker expression,

as well as S-100 protein. In difficult cases, the detection of *SYT* gene rearrangement *t(x;18)* can help the distinction. Overall, to confirm the diagnosis of GNET, the *EWSR1* gene status must be investigated [16].

CCS GIT instead deserves special consideration because it shares many molecular and genetic features with GNET. It is a rare and aggressive sarcoma that rarely occurs as a primary tumor in the gastrointestinal tract; it was considered the gastrointestinal location of a CCS tumor, an aggressive sarcoma first described by Enzinger in 1965 [17] as an uncommon sarcoma that typically arises in association with the tendons and aponeuroses in the lower limbs of young adults. CCS GIT was first reported by Ekfors et al. in 1993 [18] and later by Zambrano et al., who, to our knowledge, reported six cases of an 'osteoclast-rich tumor of the gastrointestinal tract with features similar to the CCS of soft tissue [4]. Therefore, the relationship between these two kinds of tumors is very close, which is important for the birth of the tumor's entity as GNET. Recently, Green et al. reported in literature a review showing the relationship between CCS GIT and GNET through an analysis of the clinicopathological features and the biological behavior of 58 GNET and 13 CCS GIT cases. The nature of GNET and CCS GIT is still under discussion, yielding two main hypotheses: (1) they represent distinct entities; (2) 'they are related tumors at different ends of the same morphological spectrum'. According to this study, GNET and CCS GIT may derive from a common precursor cell such as an autonomic nervous system-related primitive cell originating from a neural crest in the gastrointestinal tract with primitive/neuroectodermal/neural features, although this can be differentiated in CCS GIT along melanogenetic lines. Nevertheless, CCS GIT is characterized by distinctive histological features, immunohistochemical expression of S-100, SOX10, and melanocytic markers, and recurrent balanced translocations involving *EWSR* and *ATF1/CREB1* genes. The most helpful distinguishing feature of GNET is the presence of osteoclast-like multinucleated giant cells, which are focal and unevenly distributed, differing from the wreath-like tumor giant cells of the CCS GIT [19]. Despite the common perception that GNET is associated with a more aggressive clinical course, there is no significant difference in their biological behavior, although both clearly share a bleak prognosis.

According to this literature review, we can conclude that it is not easy to diagnose GNET because it shares morphological features with other previously described tumors, which can easily lead to an incorrect diagnosis and to a wrong therapeutical approach.

The most common treatment for patients with GNET is excision of the tumor [1]; in the case of liver metastases, as reported previously, radiofrequency and tumor resection can prolong the prognosis. A lymphadenectomy, if possible, can be an effective solution. Radiotherapy and conventional chemotherapy are not very effective in the treatment of GNET. At follow-up, they usually develop regional lymph nodes and liver metastases. In the literature, previous studies showed a poor prognosis, with the prognosis improving over time (despite a lack of data on the 5 year survival rate), probably thanks to the therapeutical approach; more experience with these rare tumors is necessary to determine the optimal treatment strategy [16]. It is also important to establish an early diagnosis of GNET because of its variable prognosis and understand the biology of GNET for helping physicians to individualize the treatment strategy [20].

**Table 1.** GNET differential diagnosis: −, negative; +, positive; Syn, Synaptophysin; ChrA, chromogranin A; NSE, neuron-specific enolase; Tyr, tyrosinase.

| Type of Tumor | Tumor Site | Symptoms | Histological Features | Immunohistochemical | Genetic Features |
|---|---|---|---|---|---|
| GIST [11,12] | Gastrointestinal tract | Asymptomatic, abdominal pain, bleeding, sign of obstruction | Spindle and sometimes epithelioid cells | c-Kit + <br> CD 34 +/− <br> DOG1 + <br> Actine + <br> Desmine − <br> S-100 − | *EWSR1* − |

**Table 1.** *Cont.*

| Type of Tumor | Tumor Site | Symptoms | Histological Features | Immunohistochemical | Genetic Features |
|---|---|---|---|---|---|
| MPNST [12] | Nerve related history of neurofi-broma/schwannoma | No gastrointestinal pain | Spindle cells/epithelioid cells | S-100 +<br>Leu-7 +<br>PGP 9,5 +<br>m.b.p. +<br>Melan A −<br>HMB45 − | *EWSR1 −* |
| GNET [12] | Gastrointestinal tract | Abdominal pain, intestinal obstruction | Proliferation of uniform spindle cells, osteoclast like multinucleated cells + | S-100 +++<br>SOX10, NB84 +/−, CD56 +/−<br>NSE +/−, Tyr −,<br>Syn +/−, HMB45 −,<br>MiTf −, MelanA − | *EWSR1 +* |
| Paraganglioma [13,14] | Sympathetic paraganglia chains, liver, lung | Sweating, diarrhea, anxiety, intermittent hypertension | Polygonal cells, | S-100 +<br>Syn +<br>NSE +<br>Chromogranin A + | *EWSR1 −* |
| NET [13,14] | Gastrointestinal tract, lung, adrenal gland | Asymptomatic, carcinoid' s syndrome | Spindle shape +/− | S-100 −<br>Syn +<br>CD56 +/−<br>Chr A + | *EWSR1 −* |
| SS [16] | Soft tissue, muscle, ligaments | Asymptomatic mass | Sheets of spindle cells | S-100 +/−<br>Epithelial markers + | *t(X;18) +*<br>*EWSR 1 −* |
| MM [16] | Lymph node, lung, liver, bones, brain | Hardened lumps under the skin, painful lymph nodes | Sheets of spindle cells | S-100 +<br>HMB-45 +, MART1 +,<br>MITF+ | *EWSR1 −* |
| CCS GIT [19] | Gastrointestinal tract | Abdominal pain, bowel obstruction | Uniform polygonal, epithelioid spindle cells | S-100 +<br>Melan A +<br>HMB45 +<br>MiTF-1 + | *EWSR1 +* |

## 4. Conclusions

In conclusion, GNET is a rare mesenchymal malignancy which occurs mainly in young and middle-aged adults, which should be considered in the differential diagnosis of gastrointestinal tract neoplasms including epithelioid cells arranged in various patterns and/or sheets of spindle cell tumor cells. Positive S-100 and SOX10 expression and negative epithelioid and melanocytic marker expression on immunostaining are helpful for diagnosis. Molecular detection of the involvement of EWSR1 chromosomal rearrangement is recommended to confirm the diagnosis of GNET [9]. Unfortunately, this kind of tumor has a poor prognosis; in fact, it shows an aggressive clinical course with local recurrence in the short term and metastasis to lymph nodes [12]. The information provided by the studies reported in the literature is helpful for a correct diagnosis, treatment, and prognosis [1,9], and further experience with these rare tumors is needed to determine the optimal therapeutical strategy [16] because there is currently no established procedure for its management [20].

**Author Contributions:** Conceptualization, C.B.; methodology, N.Z.; validation, V.C. and G.G.N.; resources, A.F.; data and image curation, M.C.; writing—original draft preparation, C.B.; writing—review and editing, N.Z.; visualization, M.C.; anatomic histological supervision, A.F. and T.F.; FISH analysys, S.C.; global supervision, V.C. and G.G.N. All authors read and agreed to the published version of the manuscript.

**Funding:** This study was not funded.

**Institutional Review Board Statement:** Not applicable.

**Informed Consent Statement:** Patient consent for publication of clinical details was obtained.

**Data Availability Statement:** Data are contained within the article.

**Acknowledgments:** The authors wish to thank Maria Di Matteo for her invaluable support in English editing.

**Conflicts of Interest:** The authors declare no conflict of interest.

**Abbreviations**

| | |
|---|---|
| GNET | gastrointestinal neuroectodermal tumor |
| CCSLGT | clear-cell sarcoma-like gastrointestinal tumor |
| ED | emergency department |
| CCS-GI | clear-cell sarcoma gastrointestinal tract |
| OLGCs | nontumoral osteoclast-like giant cells |
| FISH | fluorescence in situ hybridization |
| NET | neuroendocrine tumor |
| MPNST | malignant peripheral nerve sheets tumor |
| SS | synovial sarcoma |
| CT | computer tomography |

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
