# Peer review of "Malignant Gastrointestinal Neuroectodermal Tumor: A Case Report and Literary Review for a Rare Differential Diagnosis"

_2673-4095, doi:10.3390/surgeries4020024_

Round 1

Reviewer 1 Report

surgeries-2337827

Baccaro et al. present an extremely rare neoplasm, malignant gastrointestinal neuroectodermal tumor (GNET) in the ileum, which was first described by Zambrano in 2003 as clear cell sarcoma. In spite of clear cell sarcoma, GNET has giant osteoclast-like cells and shows diffuse and intense positivity for S-100 with no immunohistochemical melanocyte differentiation. The diagnosis is based on clinicopathological, immunophenotypic, and molecular pathology features.  

This is an interesting case report. The manuscript is well-written and is easy to read. However, a major revision is needed for publication. I would like to suggest the following points for improvement of the manuscript.

(1)  In the abstract: Please define “ED”.

(2)  Figures 1-4: There are similar figures and some arrows are floating in midair. Please use important figures for diagnosis.

(3)  Figures 6-8: All figures are taken from H&E-stained sections. Immunohistochemical photos (S100, SOX-10, CD56, Ki067, HMB-45, Melan-A, DOG-1) together with EWSR1 FISH translocation should be presented.

(4)  Additionally, a photo of osteoclast-like multinucleated giant cells can be presented.

(5)  Important papers (PMID: 36980439, PMID: 35200608, PMID: 25364450, PMID: 30930990) should be cited and discussed.

(6)  English needs editing.

English needs editing.

Reviewer 2 Report

A case report by Baccaro et al on malignant gastrointestinal neuroectodermal tumor (GNET), which is a rare soft tissue sarcoma. The introduction is sufficient, and description of the case clear enough. 

Specific comments:

1. Please perform English and typos editing e.g., line 42: "caratteristic".

2. Gene names should be written in italics.

3. Fig. 1 & 2: What is shown by the arrow in the panel b? In addition, please draw more "professional" arrows in all figures.

4. It is not necessary to split IHC figures to many separate files. Also, please revise figure descriptions.

minor revision required

Round 2

Reviewer 1 Report

The revised manuscript has greatly been improved.